# Exploring the Relationship between Food Addiction, Overweight, Obesity, and Telomere Length

**Trina Aguirre** [1,*] and **Kosuke Niitsu** [2]

1   College of Nursing-West Nebraska Division, University of Nebraska Medical Center, 1601 East 27th St., Scottsbluff, NE 69361, USA
2   School of Nursing and Health Studies, University of Washington Bothell, Bothell, WA 98011, USA; kniitsu@uw.edu
*   Correspondence: taguirre@unmc.edu; Tel.: +1-308-632-0412; Fax: +1-308-632-0415

**Abstract:** Background: Individuals with food addiction (FA) compose a distinct subset of people with obesity who are less responsive to weight loss interventions. An emerging field of study explores the role of telomere length in disease processes. Some evidence suggests that obesity is associated with telomere shortening; however, we are not aware of studies examining telomere length in obesity subtypes. Therefore, we explored whether FA and levels of obesity were associated with telomere shortening. Methods: We enrolled 120 adults (aged 19–70) with overweight/obesity (BMI ≥ 25); half were positive for severe food addiction (FA+), and half were negative for food addiction (FA−) (Yale Food Addiction Scale 2.0). Participants completed a demographic form and provided a saliva sample (Oragene saliva DNA collection kit). Telomere length was analyzed using the monoplex quantitative polymerase chain reaction (qPCR). Data were analyzed using descriptive statistics, $t$-tests, and ANOVAs ($\alpha = 0.05$). Results: Participants with overweight (mean = 1.40 t/s, SD = 0.40) had longer telomeres ($p = 0.013$) than those with morbid obesity (mean = 1.15 t/s, SD = 1.29). Telomere length did not differ ($p = 0.306$) between persons who were FA− (mean = 1.26 t/s, SD = 0.26) and those who were FA+ (mean = 1.32 t/s, SD = 0.34). The youngest participants (mean = 1.39 t/s, SD = 0.33) had longer telomeres ($p = 0.006$) than the oldest participants (mean = 1.18 t/s, SD = 0.19). Conclusion: Those who were morbidly obese had the shortest telomere lengths. Interestingly, however, there were not significant telomere length differences in the food addicted vs. nonfood-addicted subtypes.

**Keywords:** food addiction; obesity; telomeres

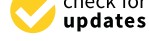



## 1. Introduction and Background

Overweight/obesity and its associated comorbidities continue to be major health concerns globally. Evidence is growing that individuals with food addiction (as determined using the Yale Food Addiction Scale 2.0 (YFAS 2.0) [1]) compose a distinct subset of people with obesity [2]. Research regarding the treatment implications of food addiction is limited, with some suggesting that individuals with food addiction may be less responsive to weight loss interventions [3]. Few studies have explored interventions to reduce food addiction, and most are preliminary [4]. Research is needed to better understand the food addiction subtype and its implications for treating obesity.

Researchers have begun investigating possible genetic relationships with food addiction (see [5] for an overview). A recent genome-wide association study identified two GW-significant loci (rs75038630 and 74902201) associated with food addiction but found only limited support for shared genetic pathways for food addiction and drug addiction [6]. An emerging field of study explores the role of telomere length in disease processes. Telomeres are the caps on the ends of chromosomes that protect them from attrition and damage [7]. Telomere shortening occurs naturally with aging but can be affected by physical activity, nutrition, tobacco use [8–11], stress [12], alcohol addiction [13],

and psychiatric disorders [14,15]. "Telomeric length serves as a mitotic clock, activating senescence and cellular cycle arrest when it reaches a shortening limit, which causes aging. Lifestyle is a factor that can affect telomeric shortening. Unhealthy habits have been linked to accelerated telomeric shortening, while healthy lifestyles are known to reduce this process and slow down aging" [16]. "Epidemiologic studies have shown that LTL predicts cardiovascular disease, all-cause mortality, and death from vascular causes" [17].

Understanding factors that contribute to telomere shortening and identifying interventions to reduce or reverse this process may help prevent or treat some health conditions. There are indications that obesity is associated with telomere shortening [18], although evidence is mixed [19]. We are not aware of any studies that examine whether telomere length differs between obesity subtypes (degrees of overweight and obesity and food addiction). As telomere shortening has been associated with alcohol addiction [13], we conducted this pilot study to explore whether telomere shortening may also be associated with food addiction (FA) and degree of overweight. We hypothesized that telomere shortening is greater in individuals with obesity who are positive for food addiction (FA+) compared to those who are negative for food addiction (FA−), as diagnosed using the YFAS 2.0 [1]. We also examined the relationship between the levels of BMI-defined obesity (overweight (25–29.9 kg/m$^2$), obesity 1 (30–34.9 kg/m$^2$), obesity 2 (35–39.9 kg/m$^2$), obesity 3 ($\geq$40 kg/m$^2$)) and telomere length. Understanding the relationship between telomere length and obesity subtypes will help us better characterize persons in each obesity subtype, which may enable us to develop interventions specific for each group. This study will also contribute to the knowledge base regarding the role of telomere length in disease processes.

## 2. Materials and Methods

### 2.1. Sample Size

The sample size for this pilot study (*n* = 120, 60 per group) was based on Darrow's [15] meta-analysis showing associations between psychiatric disorders and telomere length. Of the 32 studies included in the meta-analysis, 18 had $\leq$60 participants in one or both groups and were able to obtain significant results. Results from this pilot study will be used to determine power for future studies.

### 2.2. Participant Recruitment and Characteristics

This study was conducted in accordance with the University of Nebraska Medical Center approved IRB #0122-21-EP protocols. Participants were recruited through health fairs and snowball sampling from a rural Midwestern community.

Inclusion/Exclusion Criteria. Potential participants (adults with overweight/obesity, BMI $\geq$ 25 kg/m$^2$) were screened for FA status using the YFAS 2.0 [1]. Only those who were positive for severe FA ($\geq$6 symptoms with clinical significance) and those who were negative for FA ($\leq$1 symptom and no clinical significance) were included in the study. Anyone with active cancer or liver disease was excluded. Persons meeting eligibility criteria were given written and verbal information about the study and the opportunity to ask questions. Those choosing to participate provided written consent and were assigned an ID number to maintain confidentiality. Human subject safety plans were followed throughout the study. Participants self-reported the presence/absence of select health-related conditions/behaviors on a brief demographic form. These included their smoking, stress, hypertension, diabetes, COPD, and alcohol use status. We did not collect information on participants' use of anti-obesity medications, participation in lifestyle interventions, or whether they had undergone bariatric/metabolic surgery.

All those enrolled (*n* = 120) were adults (age 19–70) with overweight/obesity (BMI $\geq$ 25 kg/m$^2$). Half were positive for severe FA ($\geq$6 symptoms with clinical significance), and half were negative for FA ($\leq$1 symptom and no clinical significance) (the control group). These criteria were selected to best define the FA+ and FA− subtypes [2]. Most participants were female (87.5%) and Caucasian (65%). The remainder were Latino

(24%), African American (7%), Asian (4%), or Native American (2%). Reported health conditions include diabetes ($n = 6$), hypertension ($n = 19$), and smoking ($n = 4$).

### 2.3. Saliva Sample Collection

Following the informed consent process, each participant received a short demographic form and an Oragene saliva DNA collection kit (Genotek Inc., Ottawa, ON, Canada) with instructions for collecting a saliva sample. A 2 mL sample of liquid saliva was obtained by participants spitting into the Oragene saliva DNA collection kit containing 1 mL of DNA stabilizing liquid. The PI (TA) collected the samples from the participants during the hours of 8:00 a.m.–4:00 p.m. and at least one hour after any food or drink. Samples were securely stored at room temperature in a locked file cabinet until all samples were collected. Per Oragene guidelines, saliva samples are stable at room temperature for up to 5 years. Our samples were collected and analyzed within 5 months.

### 2.4. Telomere Measurement

The samples, identified only by their ID numbers (to reduce risk of bias), were then shipped to the Elizabeth Blackburn Laboratory at the University of California, San Francisco and stored at room temperature until processing for DNA extraction. They were analyzed for telomere length using the monoplex quantitative polymerase chain reaction (qPCR) method. This method has been validated against the Southern blot method. Stout et al. [20] ran the inter-assay variability test and found that the average CV was 2.7% for saliva DNA. Both the saliva collection kit and the telomere length measurement were used only for research purposes. Stout et al. [20] reported that salivary telomere length was correlated with whole blood telomere length (r = 0.56, $p = 0.005$); given limits with minimally invasive sampling techniques, measuring TL from the dried blood spot may demonstrate a better external validity than saliva [21]. Laboratory personnel were blinded to the obesity subtype of the persons providing the samples. Once analyzed, the samples were destroyed via autoclave at the laboratory facility.

### 2.5. Data Analysis

We evaluated whether telomere length differed between individuals with obesity who were severe FA+ and those who were FA− and explored differences in telomere length based on demographic characteristics and obesity level. Data were analyzed using descriptive statistics and two-tailed *t*-tests and ANOVAs. Bonferroni tests were used to separate means for significant ANOVA results. Kolmogorov–Smirnov tests were used to evaluate normality. Statistical analyses were performed using IBM SPSS Statistics Version 28.0.0.0 (190). Significance was determined based on $\alpha = 0.05$.

## 3. Results

### 3.1. Biometric and Demographic Variables

One hundred twenty samples were submitted for analysis; two returned no DNA. Thus, statistical analyses were conducted using data from 58 FA− and 60 severe FA+ persons. One-way ANOVAs revealed differences in telomere length based on some of participants' biometric and demographic characteristics. The most interesting finding was that telomere length tended to shorten with increasing obesity, with participants in the overweight category (mean = 1.40 t/s, SD = 0.40) having longer telomeres ($p = 0.013$) than those in the morbidly obese category (mean = 1.15 t/s, SD = 1.29) (Table 1). Consistent with previous studies, [22–24], those in the youngest age category (mean = 1.39 t/s, SD = 0.33) had longer telomeres ($p = 0.006$) than those in the oldest age category (mean = 1.18, SD = 0.19) (Table 1). Based on race/ethnicity, non-Hispanic White persons (mean = 1.25 t/s, SD = 0.28) had shorter telomere length ($p = 0.007$) than African Americans (mean = 1.67 t/s, SD = 0.67) (Table 1).

**Table 1.** Comparison of telomere length (t/s) based on demographic and health-related variables using two-tailed independent *t*-tests or ANOVAs and Bonferroni tests ($\alpha$ = 0.05).

| Category | N | Mean [1] | Std Dev | df | *p* |
|---|---|---|---|---|---|
| BMI (kg/m$^2$) | | | | | |
| Overweight (25–29.9) | 36 | 1.39956 * | 0.404430 | 3 | 0.013 |
| Obesity 1 (30–34.9) | 40 | 1.30955 | 0.259616 | | |
| Obesity 2 (35–39.9) | 20 | 1.22735 | 0.241782 | | |
| Obesity 3 ($\geq$40) | 22 | 1.14786 * | 1.29293 | | |
| Age (years) | | | | | |
| 19–30 | 48 | 1.39313 ** | 0.332638 | 2 | 0.006 |
| 31–50 | 42 | 1.25607 | 0.299637 | | |
| 51–70 | 28 | 1.17646 ** | 0.185458 | | |
| Race/Ethnicity | | | | | |
| Non-Hispanic White | 76 | 1.24505 ** | 0.275662 | 3 | 0.007 |
| Hispanic White | 28 | 1.33379 | 0.211249 | | |
| African American | 6 | 1.67167 ** | 0.669976 | | |
| Other | 8 | 1.32075 | 0.265295 | | |
| Gender | | | | | |
| Female | 104 | 1.28917 | 0.315355 | 116 | 0.715 |
| Male | 14 | 1.32086 | 0.190804 | | |
| Smoking | | | | | |
| No | 114 | 1.29850 | 0.305689 | 116 | 0.288 |
| Yes | 4 | 1.13425 | 0.147536 | | |
| Stress | | | | | |
| No | 115 | 1.29361 | 0.306212 | 116 | 0.881 |
| Yes | 3 | 1.26700 | 0.137011 | | |
| Hypertension | | | | | |
| No | 99 | 1.30290 | 0.319214 | 116 | 0.417 |
| Yes | 19 | 1.24100 | 0.194636 | | |
| Diabetes | | | | | |
| No | 112 | 1.29871 | 0.307968 | 116 | 0.373 |
| Yes | 6 | 1.18517 | 0.159885 | | |
| COPD | | | | | |
| No | 117 | 1.29444 | 0.303680 | 116 | 0.562 |
| Yes | 1 | 1.11700 | --- | | |
| Alcohol Use (drinks per day) | | | | | |
| 0 | 100 | 1.28220 | 0.271574 | 3 | 0.736 |
| 1 | 16 | 1.34612 | 0.469677 | | |
| 2 | 1 | 1.54000 | --- | | |
| $\geq$4 | 1 | 1.26800 | --- | | |
| Food Addiction Status | | | | | |
| FA− | 58 | 1.26376 | 0.259768 | 116 | 0.306 |
| Severe FA+ | 60 | 1.32113 | 0.339129 | | |

[1] One asterisk designates means that differed at $\alpha$ = 0.05; double asterisks designate means that differed at $\alpha$ = 0.01.

Telomere length did not differ for other demographic and health-related variables (independent *t*-tests; *p* = 0.288–0.881). These include participants' gender and smoking,

stress, hypertension, diabetes, COPD, and alcohol use status (Table 1). The small sample sizes for these variables limited our ability to discern differences and make inferences.

*3.2. Food Addiction Status*

Based on an independent *t*-test, telomere length did not differ (*p* = 0.306) between persons with overweight/obesity who were FA− (mean = 1.26 t/s, SD = 0.26) and those who were severe FA+ (mean = 1.32 t/s, SD = 0.34) (Table 1).

## 4. Discussion

Telomere length was shortest in persons with morbid obesity (obesity 3; BMI $\geq$ 40 kg/m$^2$) and longest in those with overweight (BMI 25–29.9 kg/m$^2$). This supports the conclusion that obesity may play a role in telomere shortening and indicates that the level of obesity matters. The earlier that intervention can be implemented to reduce overweight/obesity, the better.

That no difference in telomere length was observed between individuals who were severe FA+ and FA− suggests that food addiction did not further shorten telomere length beyond that associated with participants being overweight/obese.

Although few African Americans participated in this study (3 severe FA+, 3 FA−), their telomere length was longer than those of non-Hispanic whites. [24] reported that the association between telomere length and all-cause mortality varied with age and race/ethnicity. They found that increasing telomere length was associated with a lower risk of dying (all-cause mortality) for Blacks $\geq$ 45 years of age and non-Hispanic Whites $\geq$ 65 years of age. Our pilot study focused on comparing telomere length in obesity subtypes rather than among racial/ethnic groups; therefore, recruitment focused on participants' FA status and level of obesity. The limited diversity of our sample limits our inferences regarding race/ethnicity.

Our results are consistent with those of Gampawar et al.'s meta-analysis [22] in finding that BMI and age are inversely associated with telomere length. That we did not observe differences in telomere length based on participants' gender and smoking, stress, hypertension, diabetes, COPD, and alcohol use status may, in part, reflect that few men participated in the study and few participants reported having these health-related conditions or behaviors (e.g., 88% of participants were female and only 16% had hypertension). Although others have found an association between stress and telomere shortening [23], we did not; this is likely because only three participants (3% of this sample) reported having high stress.

Limitations: The focus of this pilot study was to explore differences in telomere length among persons in different obesity and FA subgroups; therefore, persons with healthy BMIs were not included which limits our inferences. As is common in obesity research, males were underrepresented in this pilot. Future larger-scale studies will need targeted recruitment efforts to increase participation by men. The small number of participants reporting smoking, stress, hypertension, diabetes, COPD, and alcohol use limited our ability to discern differences and make inferences about these variables. Larger sample sizes will be needed to more definitively explore any effects associated with these health-related conditions/behaviors and any confounding variables that were not evaluated in this pilot study. The cross-sectional nature of this study is also a limitation. That all participants were recruited on a rural Midwestern college campus limits the generalizability of our findings.

## 5. Conclusions

Findings: Participants with overweight had longer telomeres than those with morbid obesity. Telomere length did not differ between persons who were FA− and those who were severe FA+. The youngest participants had longer telomeres than the oldest.

Significance: That telomere length was shortest in those with morbid obesity supports a decline in health with increasing obesity. The lack of difference in telomere length between FA− and severe FA+ participants indicates that more study is needed to discern the impact

of food addiction on the health of persons with overweight/obesity and its implications for treating overweight/obesity.

Implications forTreatment of Overweight/Obesity: Much is yet to be learned about the role of telomere length in disease processes and the factors that influence telomere length. The decline in telomere length with increasing obesity (overweight vs. morbid obesity) suggests that health care professionals should intervene early to prevent patients from becoming morbidly obese. There is growing evidence that some interventions (e.g., healthy diet, physical activity) may alter telomere length [25,26]. How food addiction might affect such interventions is unknown. That food addiction is associated with less success in weight loss/maintenance interventions [2], suggests that this is an area that should be further explored.

**Funding:** The authors disclosed receipt of the following financial support for the research, authorship, and/or publication of this article. This work was supported by the UNMC Diversity Grant [Internal, 2019–2022]; the Joan McVay Grant [No number, 2021]; and the University of Washington Bothell School of Nursing & Health Studies Pilot Grant [No number, 2021].

**Institutional Review Board Statement:** EXPEDITED REVIEW CATEGORY: 45 CFR 46.110, Category 3. Prospective collection of biological samples for research purposes by noninvasive means. #0122-21-EP.

**Informed Consent Statement:** Informed consent was obtained from all subjects involved in the study.

**Data Availability Statement:** ClinicalTrials.gov.

**Acknowledgments:** Thank you to Ann Koehler, research assistant/editor, for her assistance with the manuscript. We used the STROBE cross sectional checklist when writing our report [von Elm E, Altman DG, Egger M, Pocock SJ, Gotzsche PC, Vandenbroucke JP]. The Strengthening the Reporting of Observational Studies in Epidemiology (STROBE) Statement: guidelines for reporting observational studies.

**Conflicts of Interest:** The authors declare no conflicts of interest.

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
