# Peer review of "Exploring the Relationship between Food Addiction, Overweight, Obesity, and Telomere Length"

_2673-4168, doi:10.3390/obesities4020007_

Round 1

Reviewer 1 Report

Comments and Suggestions for Authors

Manuscript ID: obesities-2919158

Title: "Exploring the Relationship Between Food Addiction, Overweight, Obesity and Telomere Length"

Authors:  Trina Aguirre et al.

The authors in the present study explored the association between food addiction (FA), obesity levels, and telomere shortening. They recruited 120 adults classified as overweight or obese and categorized them based on their FA status. Telomere length was measured using quantitative polymerase chain reaction (qPCR). The results indicated that individuals with morbid obesity exhibited significantly shorter telomeres than those classified as overweight. However, telomere length did not significantly differ between individuals with and without FA. The following points merit consideration.

Comments:

1.     The conclusions presented in the abstract need to be revised to accurately summarize and reflect the study's findings. I suggest avoiding phrases such as "sadly" and refraining from mentioning references or statements about children and well-known facts from previous studies. Instead, if such information is deemed necessary, it should be integrated into the background section to support the formulation of the study hypothesis.

2.     Regarding the power analysis, I suggest providing more comprehensive details on the power calculation performed, including the effect size, the desired power level (e.g., 0.80), the significance criterion (alpha level), consideration of potential dropout rates, and the specific statistical method or software used. This information would greatly enhance the transparency and reproducibility of your study's methodology.

3.     Please specify the inclusion and exclusion criteria for the study.

4.     Did the authors consider the importance of including a control group of normal-weight individuals?

5.     Clarify if participants were using anti-obesity medications (e.g., GLP-1 receptor agonists, etc.), engaging in lifestyle interventions, or had undergone bariatric/metabolic surgery.

6.     Could you briefly describe how the samples were selected and stored?

7.     In your statistical analysis section, include additional details regarding normality assessment. Also, specify whether two-tailed tests were used and confirm that post-hoc tests were conducted following significant ANOVA results.

8.     Please provide a brief overview of the FA assessment methods and tools employed. The term “60 severe FA+” is mentioned (line 122). Could you define what constitutes severe FA?

9.     It is suggested to include a detailed table describing the demographic characteristics, clinical characteristics, medical conditions, and treatments of the study participants for the total population and the main study subgroups. Indicate whether these variables differ significantly among the groups. If space is limited, consider using symbols to denote significance levels and the degree of significance between groups.

10.  Consider performing a multivariable analysis to identify potential variables associated with telomere length. Incorporating such analysis could enhance the significance of the findings.

11.  I recommend extending the section on limitations to cover all potential limitations of the study. Summarize these limitations in a separate paragraph within the discussion section to provide a comprehensive view of the study's constraints.

12.  Please clarify the rationale behind focusing exclusively on nursing practice ("Implications for Nursing Practice" section), rather than encompassing the broader clinical management of patients with overweight or obesity. Expanding the scope of implications to reflect the complexity of these conditions could make the findings more relevant to a wider audience, thereby amplifying the impact of the research.

13.  Please add a conclusion paragraph at the end of the main text to succinctly summarize the study's findings, significance, and potential implications for practice and future research.

Author Response

  1. Abstract -avoiding phrases such as "sadly" and refraining from mentioning references or statements about children and well-known facts from previous studies. Instead, if such information is deemed necessary, it should be integrated into the background section to support the formulation of the study hypothesis- These sentences were removed from the abstract.
  2. providing more comprehensive details on the power calculation performed, including the effect size, the desired power level (e.g., 0.80), the significance criterion (alpha level), consideration of potential dropout rates, and the specific statistical method or software used. 

    For this pilot study we used Darrow's meta-analysis. Data from this exploratory study will be used to power future work.The sample size for this pilot study (n = 120, 60 per group) was based on Darrow’s (2016) meta-analysis showing associations between psychiatric disorders and telomere length. Of the 32 studies included in the meta-analysis, 18 had ≤ 60 participants in one or both groups and were able to obtain significant results.

  3. Inclusion/Exclusion criteria-The terms inclusion/exclusion criteria were added to the method section and overweight/obesity were defined by BMI.
  4. Control group?  -FA- participants were the control group. Those terms were added. Normal weight persons were irrelevant to this study since the focus of this study is for potential differences in treatments for overweight/obese persons based on presence or absence of food addiction
  5. Clarify variables that could affect outcomes.-Cancer and liver disease are most likely to adversely affect telomere length. Therefore were used as exclusion criteria. We also examined. Other factors that may influence telomere length were gender, smoking, stress, hypertension, diabetes, COPD and alcohol use.  We found no significant relationships between telomere length and these variables. However, the sample sizes with these variables were very small.

  6. Describe sample collection  and storage-this section - Added more detail in the sample collection section.
  7. Statistical section-Added to this section" that two tailed tests were used and post hoc tests were conducted following significant ANOVA results."
  8. Define severe FA- Defined in section 2.2. Six or more clinical symptoms.
  9. demographics, clinical conditions, treatments, medical conditions-For demographics only race and gender were collected for this exploratory pilot study. Only medical or clinical conditions known or suspected to affect telomere length were collected and noted in the methods.
  10. multivariable analysis-Sample sizes were too small for multivariate analysis to establish significance.
  11. limitations-Completed a new limitations section
  12. consider implications for all professions- Yes noted and changed in Conclusion section.
  13. provide conclusion of findings, significance and implications for future practice. Added a conclusions section with findings, significance and implications section.

Reviewer 2 Report

Comments and Suggestions for Authors

1. Introduction and Background

·         The authors used relevant literature to situate their investigation in the research landscape. References to research on food addiction, obesity, and telomere length form a strong basis. However, the section may benefit from a more in-depth review of past results on telomere length in various obesity subtypes, even though the authors acknowledge a scarcity of such research.

·         The introduction presents the study's hypothesis on telomere length disparities across persons with and without FA and throughout obesity levels. These hypotheses are reasonable and anchored in the presented context; however, a more explicit link between the hypotheses and the theoretical framework would enrich this section.

·         Terms like "food addiction," "telomere length," and "obesity subtypes" are used consistently. Still, the manuscript would benefit from more precise definitions of these key concepts, especially how "food addiction" is operationalized in this study.

·         The authors articulate the potential contributions of their study to the field, specifically regarding interventions for obesity subtypes. However, elaborating on how this research might inform specific clinical practices or policy changes could enhance this section.

2. Materials and Method

Power Analysis (2.1):

·         The paper cites a previous meta-analysis to determine the sample size, which is a valid method. However, the statistical parameters utilized in the power analysis (effect size, power, and alpha level) are not specified. Detailing these factors would improve the section and provide a more complete understanding. The decision to include 60 participants per group should be supported by a particular justification relevant to the setting of this study, taking into account its unique goals and demographics.

2. Participant Recruitment and Characteristics (2.2):

·         The recruiting approach of health fairs and snowball sampling in a rural Midwest town is deemed adequate. However, the study's generalizability may be restricted due to the sampling procedure and population demographics. More information on how the snowball sampling was carried out and how bias was reduced during the procedure would be useful. Furthermore, the high proportion of female participants (87.5%) raises concerns about the study's relevance to a larger population, and this gender imbalance should be addressed in the limitations.

3. Saliva Sample Collection (2.3):

·         The method is well outlined. However, including any steps to maintain consistency in the sample collection schedule would be useful, as diurnal fluctuations may alter biomarker levels.

4. Telomere Measurement (2.4):

·         The qPCR technique for measuring telomere length is fully detailed. The validation against the Southern blot method and the inter-assay variability check are critical aspects that provide credibility to the approach. However, highlighting the limitations of the qPCR approach in telomere length assessment, particularly its relative quantitative nature, would offer a more balanced perspective.

3. Results

3.1. Biometric and Demographic Variables

·         This section summarizes the findings for biometric and demographic variables, focusing on the association between telomere length and parameters such as obesity level, age, and race/ethnicity. The finding that telomere length decreases with increasing obesity levels is corroborated by statistical data (P=.013 when comparing overweight to morbidly obese). The findings are consistent with current research, which strengthens the case. However, the section may benefit from a more nuanced assessment of these findings' statistical significance and potential therapeutic importance. For example, while the study indicates a substantial variation in telomere length between obese categories, it does not investigate these disparities' molecular or clinical significance. Briefly describing how these telomere length differences may affect health outcomes or dangers would offer a more complete picture of the study's consequences.

·         The section also mentions that telomere length did not differ substantially across other demographic characteristics, including gender, smoking status, and other health issues. Given the complexities of telomere biology, it would be useful to frame these null findings within the larger research landscape, recognizing any limits or confounding variables that may have impacted the results.

3.2. Food Addiction Status

·         The analysis regarding food addiction status is concise and clearly stated. The lack of significant difference in telomere length between the FA- and FA+ groups (P=.306) is an interesting result, suggesting that food addiction status alone, as defined in this study, may not be associated with telomere length among individuals with overweight or obesity. However, this section could be enhanced by addressing potential reasons for the absence of a significant difference and exploring the implications of this finding within the existing body of research. For instance, considering the multifactorial nature of telomere length regulation, it would be insightful to discuss how other factors (e.g., diet quality, physical activity levels, psychological stress) might interact with food addiction to influence telomere length.

4. Discussion

·         This well-organized discussion relates study findings to broader research concerns. Given the lack of substantial telomere length modifications across states linked to food addiction, a more detailed review of how these data fit into the research body would benefit this area.

·         The authors emphasize the need for early intervention by suggesting that obesity contributes to telomere shortening. Although this is the case, a more detailed explanation of the results will be provided by going into the several ways that obesity influences telomere length in this section.

The authors acknowledge the lack of individuals with healthy BMI for comparison. To provide a more balanced picture, consider other limitations, like the study's cross-sectional design, unmeasured confounders, and generalizability based on the unique sample population.

·         The discussion could be improved by making specific recommendations or considerations for future research, particularly on the role of food addiction in telomere length and the larger context of obesity research.

·         The authors may suggest future research directions, like longitudinal studies demonstrating causation or intervention trials targeting food addiction to measure impacts on telomere length.

To improve the discussion, the authors should include more current research or meta-analyses that have looked at comparable themes, offering a more complete context for interpreting their findings.

Comments on the Quality of English Language

There are a few places in the text where little grammatical or stylistic changes might be made, but overall, the content is intelligible and straightforward. Nonetheless, the document's general coherence and clarity imply that the information is well-structured and the reasoning is presented logically, but it needs little modification.

Author Response

  1. Define food addiction added in background-; define telomere length-(defined in lines 45-47); define obesity subtypes (defined in background)-;elaborate on informing clinical practice- changed in conclusion
  2. 2.1Effect size, power and alpha level of power analysis; why 60 per group 2.2 demographic table; generalizability-for this exploratory pilot study we based our sample on the Darrow meta analysis. Data from this study will allow us to generate a Power analysis for future work.
  3.  2.3steps to maintain consistency in sample collection- added more information here
  4. 2.4 limitations of q PCR.-- KO to respond
  5. 3.12 Briefly describing how these telomere length differences may affect health outcomes or dangers -expanded text regarding length and health in background
  6. 3.2 discuss how other factors (e.g., diet quality, physical activity levels, psychological stress) might interact with food addiction to influence telomere length- see above
  7. 4 going into the several ways that obesity influences telomere length in this section; consider other limitations, like the study's cross-sectional design, unmeasured confounders, and generalizability based on the unique sample population; going into the several ways that obesity influences telomere length;  making specific recommendations or considerations for future research, particularly on the role of food addiction in telomere length and the larger context of obesity research; suggest future research directions, like longitudinal studies demonstrating causation or intervention trials targeting food addiction to measure impacts on telomere length.-added to limitations section
  8. Grammar-reviewed grammar

Round 2

Reviewer 1 Report

Comments and Suggestions for Authors

Manuscript ID: obesities-2919158R1

Title: "Exploring the Relationship Between Food Addiction, Overweight, Obesity and Telomere Length"

Authors:  Trina Aguirre et al.

The authors have made efforts to address some of my comments and suggestions by revising their manuscript and improving the paper further. However, several points still require clarification:

  1. Since a formal power analysis was not performed, I recommend changing the title 2.1 to 'Sample Size.'
  2. Were participants using anti-obesity medications (e.g., GLP-1 receptor agonists, etc.), involved in lifestyle interventions, or had undergone bariatric/metabolic surgery?
  3. How long were the samples stored at room temperature, and did the investigators consider the possibility of DNA degradation during this period?
  4. How was normality assessed?
  5. Adding the previously suggested table, based on the available information, would aid readers in drawing significant conclusions regarding potential differences/similarities among the groups.
